# Activation of G Protein-Coupled Estrogen Receptor (GPER) Negatively Modulates Cardiac Excitation–Contraction Coupling (ECC) through the PI3K/NOS/NO Pathway

**DOI:** 10.3390/ijms25168993

**Published:** 2024-08-19

**Authors:** Leandro A. Diaz-Zegarra, María S. Espejo, Alejandro M. Ibañez, Mónica E. Rando, Lucia E. Pagola, Verónica C. De Giusti, Ernesto A. Aiello

**Affiliations:** 1Centro de Investigaciones Cardiovasculares “Dr. Horacio E. Cingolani”, Facultad de Ciencias Médicas, Universidad Nacional de La Plata—CONICET, La Plata 1900, Buenos Aires, Argentina; leandrodiaz@ciclaplata.org.ar (L.A.D.-Z.); mse@biomed.au.dk (M.S.E.); aleibanez@med.unlp.edu.ar (A.M.I.); monicarando@ciclaplata.org.ar (M.E.R.); luciapagola@ciclaplata.org.ar (L.E.P.); vdegiusti@med.unlp.edu.ar (V.C.D.G.); 2Biomedicine Department, Health, Aarhus University, 8000 Aarhus, Midtjylland, Denmark

**Keywords:** Wistar rats, cardiomyocytes, G protein-coupled estrogen receptor (GPER), L-type calcium current, calcium transient, cell shortening

## Abstract

The G-protein-coupled estrogen receptor (GPER) has been described to exert several cardioprotective effects. However, the exact mechanism involved in cardiac protection remains unclear. The aim of this study is to investigate the role of GPER activation on excitation–contraction coupling (ECC) and the possibility that such effect participates in cardioprotection. The cardiac myocytes of male Wistar rats were isolated with a digestive buffer and loaded with Fura-2-AM for the measurement of intracellular calcium transient (CaT). Sarcomere shortening (SS) and L-type calcium current (I_CaL_) were also registered. The confocal technique was used to measure nitric oxide (NO) production in cells loaded with DAF-FM-diacetate. Cardiac myocytes exposed to 17-β-estradiol (E2, 10 nM) or G-1 (1 μM) for fifteen minutes decreased CaT, SS, and I_CaL_. These effects were prevented using G-36 (antagonist of GPER, 1 μM), L-Name (NO synthase -NOS- inhibitor, 100 nM), or wortmannin (phosphoinositide-3-kinase -PI3K- inhibitor, 100 nM). Moreover, G1 increased NO production, and this effect was abolished in the presence of wortmannin. We concluded that the selective activation of GPER with E2 or G1 in the isolated cardiac myocytes of male rats induced a negative inotropic effect due to the reduction in I_CaL_ and the decrease in CaT. Finally, the pathway that we proposed to be implicated in these effects is PI3K-NOS-NO.

## 1. Introduction

The G protein-coupled estrogen receptor (GPER) was initially named GPR30 and was characterized as an orphan receptor [1,2]. In 2006, a selective agonist of GPR30 named G1 was synthesized using molecular screening and computational testing [3]. Finally, the receptor was renamed GPER by the International Union of Basic and Clinical Pharmacology (IUPHAR) in 2007 [4,5]. As G1 is a non-steroidal compound with high selectivity to GPER and not to classic estrogen receptors (ERs) [3], it represents a useful strategy to investigate the implications of GPER activation.

It is well known that 17-β-estradiol (E2) activates complex pathways involving both genomic and non-genomic pathways within different cell types. In the heart, while the genomic mechanisms are well characterized, being mediated by the classical estrogen receptor [6,7], GPERs present non-genomics effects, either with E2 or G1 [8,9]. The selective activation of GPER triggers several intracellular pathways like extracellular signal-regulated kinases (ERKs), phosphoinositide 3-kinase (PI3K), cyclic adenosine monophosphate (cAMP), protein kinase A (PKA), endothelial nitric oxide synthase (eNOS), soluble guanylate cyclase (sGC), and protein kinase B (AKT) [8,9,10,11,12,13]. In 2015, we demonstrated that G1 binding to GPER can transactivate to the epidermal growth receptor (EGFR), leading to the activation of the PI3K-AKT pathway, which, finally, activates the electrogenic isoform of the sodium/bicarbonate cotransporter (eNBC) in isolated ventricular myocytes [14].

It was fully demonstrated that GPER activation affords cardioprotection. Chronic GPER activation decreases blood pressure [15] and prevents diastolic dysfunction [16,17] and progression to heart failure in rats treated with isoproterenol [18]. Moreover, GPER activation prevents ischemic/reperfusion injury with less myocardial inflammation [19], reducing infarct size with the activation of PI3K-AKT and extracellular signal-regulated kinase 1/2–Erk1/2-glycogen synthase kinase 3β (MEK1/2-Erk1/2-GSK-3β) pathways to inhibit mitochondrial permeability transition pore (mPTP) opening [20] and by reducing mitochondrial dysfunction, mitophagy, and regulation of the mPTP opening [21]. Accordingly, our laboratory recently demonstrated, in a rat ovariectomy (OVX) model, that the chronic administration of G1 enhanced cardiac mitochondrial function, improved cardiac function, and reduced infarct size after ischemia [22].

Interestingly, another possibility to prevent cardiac dysfunction after ischemic/reperfusion damage is L-type calcium channel (LTCC) inhibition [23,24,25]. LTCC is located in cardiac T tubules [26,27,28], and calcium influx activates the ryanodine receptors (RyRs) of the sarcoplasmic reticulum (SR), triggering the calcium-induced calcium-release (CICR) mechanism. This calcium transient is responsible for activating the contractile apparatus in the cardiomyocyte (excitation–contraction coupling, ECC). Modification in LTCC activity could induce changes in ECC, triggering pathologic events like cardiac arrhythmias, heart failure, and cardiac hypertrophy [29,30]. Several studies have demonstrated that LTCC function decreases after treatment with high concentrations of estrogen [31,32] and in the OVX model [33]. Even with the deletion of classic ERs, E2 decreases the calcium current [34], indicating that another receptor acts in the presence of E2. The activation of GPER prevents the effect of isoproterenol on calcium current with E2 or G1, decreasing it [32,35]. However, the signaling behind the effect of GPER on LTCC remains unclear.

We have previously demonstrated that GPER is present in T tubules of cardiomyocytes [36], suggesting that their signaling can be relevant to ECC. Thus, in the present study, we tested the hypothesis that the activation of GPER and its downstream intracellular pathway PI3K and NOS alters calcium handling by inhibiting LTCC and subsequently affecting ECC.

## 2. Results

### 2.1. E2 and G1 Reduce Cardiomyocyte Contractility through GPER Signaling

In order to investigate the effect of GPER on cardiac contractility, isolated ventricular myocytes were exposed to E2 (10 nmol/L) for 15 min. Figure 1A shows a representative continuous sarcomere shortening (SS) recording of one cardiomyocyte subjected to the acute application of E2 (10 nmol/L). Figure 1B depicts the average values of the negative inotropic effect of E2. Moreover, this effect was prevented when the cells were preincubated with the GPER blocker G36 (1 μmol/L, Figure 1C,D), suggesting that it is mediated by GPER activation and not by the stimulation of the canonical E2 receptors (ERs). Although we cannot confirm it, due to the rapid onset of this negative inotropic effect, it is also feasible to suggest that it is a non-genomic pathway. As shown in Figure 2A,B, the acute administration of the selective GPER agonist G1 (1 μmol/L) decreased sarcomere shortening in a similar magnitude to E2, reinforcing the idea that the activation of GPER is involved in the reduction in cardiac contractility. As expected, the effect of G1 was prevented by G36 (Figure 2C,D). To determine if the GPER-induced negative inotropic effect is a consequence of intracellular calcium (Ca^2+^_i_) reduction, we recorded calcium transients (CaT) in the presence of E2 or G1. As shown in Figure 3, CaT decreased with E2 (panel A) or G1 (panel C), and these effects were prevented when the myocytes were preincubated with G36 (panels B and D). One possibility might be that a reduction in sarcoplasmic reticulum (SR) Ca^2+^ reuptake leads to a reduction in SR Ca^2+^ content. To investigate this, a caffeine-induced calcium transient was performed at the end of each experiment. However, G1 had no significant effect on caffeine-induced calcium transient amplitude (Figure 3E). Therefore, the activation of GPER signaling reduces contractility by diminishing Ca^2+^ handling without affecting SR Ca^2+^ content.

### 2.2. Activation of GPER Decreases L-Type Calcium Current (I_CaL_) in Cardiac Myocytes

SR Ca^2+^ release through RyR induced by the Ca^2+^ influx produced by the opening of the sarcolemmal LTCC during the action potential plateau represents the basis of ECC. Thus, we employed the patch clamp technique in the whole-cell configuration to assess the participation of I_CaL_ in the GPER-induced negative inotropic effect observed in the previous experiments. E2 or G1 reduced I_CaL_ recorded at 0 mV by approximately 25% (Figure 4A and Figure 5A, respectively). This effect was prevented when the cells were preincubated with G36 (Figure 4B and Figure 5B), indicating that E2 and G1 exert their effect through the activation of GPER. The reduction in I_CaL_ induced by E2 or G1 was observed at different test voltages, as depicted in the current density vs. voltage (IV) plots of Figure 4C and Figure 5C. Overall, these data demonstrate that GPER activation reduces I_CaL_, affecting ECC and producing a negative inotropic effect in cardiac myocytes.

### 2.3. PI3K and NOS Are Involved in the GPER-Triggered Intracellular Pathway

It is known that GPER activation stimulates the PI3K and NOS pathways, modulating diverse cardiac functions [8,10,14]. Thus, we used wortmannin (100 nM) and L-Name (100 nM) to block PI3K and NOS, respectively. The inhibitors were used before and after exposing cardiomyocytes to G1 and SS, and CaT and I_CaL_ were measured. Figure 6 shows that the effect of G1 is abolished in the presence of a PI3K or NOS blockade, indicating that the GPER intracellular pathway involves PI3K and NOS activation.

Finally, to understand if the effect of GPER initiates with the activation of NOS or with the stimulation of PI3K, we measured nitric oxide (NO) production with the fluorescent dye DAF-FM-diacetate. The cells were perfused with G1 or wortmannin plus G1. As shown in Figure 7, G1 increased the production of NO. This effect is prevented when the cardiomyocytes were pretreated with wortmannin (Figure 7A,B), indicating that GPER acts by activating PI3K, which subsequently induces an increase in NO production by NOS stimulation.

## 3. Discussion

In the present paper, we demonstrated for the first time that cardiac GPER activation with E2 or G1 induced a rapid negative inotropic effect through the downregulation of LTCC, which decreases I_CaL_ and CaT, subsequently reducing cardiomyocyte contractility. In association with this mechanism, a previous report from our laboratory has demonstrated that GPER is located in T tubules of cardiomyocytes, suggesting that its signaling could be relevant for ECC [36]. Although it is not clear at this time whether this effect of GPER activation and decrease in Ca^2+^ is cardioprotective, it could be speculated that in certain conditions, for example, reperfusion following ischemia where there is a sudden intracellular Ca^2+^ overload, our proposed mechanism protects the cardiomyocyte. Accordingly, previous studies have demonstrated favorable results using different LTCC blockers [23,24,25], suggesting the important contribution of Ca^2+^ influx across these channels during reperfusion.

The Idea of using GPER as a therapeutic target to prevent the development and progression of several cardiovascular diseases is still under investigation and needs further research. Different works have demonstrated that the chronic activation of GPER produces cardiovascular protection like blood pressure reduction in an ovariectomy (OVX) model [15], improvement of heart function after ischemia/reperfusion injury, independently of animal sex [10], and prevention of diastolic dysfunction in salt-induced hypertension [16], among other publications. Consistently, we have recently reported that OVX rats exert a deteriorated mechanical response after ischemia and reperfusion injury and bigger infarct size, possibly due to an impaired mitochondrial function, which was successfully prevented with the chronic treatment with G1 [22]. Interestingly, the acute application of E2 during reperfusion after ischemia has also been demonstrated as a cardioprotective alternative [21]. In this case, the authors proposed that GPER activation, increasing NO production, and activating ERK 1/2 kinase can prevent mitochondrial transition pore aperture. This acute cardioprotective effect of GPER activation was also reported by Rocca et al. when they identified the PI3K-AKT-eNOS pathway as being able to improve mitochondrial survival [37]. Nevertheless, the GPER-induced modulation of ECC reported herein might represent a novel non-genomic cardioprotective pathway that deserves future elucidation.

Several publications have demonstrated that the activation of GPER involves the increased activity of PI3K and NOS [8,10,13,14]. To determine whether this pathway is responsible for contractility and Ca^2+^ handling, we use inhibitors for each possible candidate. When we used L-Name to inhibit NOS, GPER did not generate an effect on ECC, indicating that NOS represents one actor in GPER actions. However, since L-Name is a general blocker of NO production, we cannot elucidate which NOS is implicated in this effect. Several reports have described that the activity of Ca^2+^ channels could be modulated by NO [38,39]. Consistent with our results, α1C subunit have nitrosylation sites that negatively modulate channel activity [38]. In addition, previous reports have shown that specific inhibition of neuronal nitric oxide synthase (nNOS) induced increments of LTCC activity [40]. Moreover, transgenic mice overexpressing nNOS evidenced a negative inotropic effect associated with LTCC block [41]. However, another study described that NO donors increase calcium current [42]. One possibility to explain this controversy is that the final effect of NO on the channel depends on the dose of this gas. Nevertheless, we can speculate that the pathway involved herein could be mediated by a possible nitrosylation of the channel, with the subsequent depression of its activity. However, we cannot discard the NO-induced activation of the guanylate cyclase–cyclic guanosine monophosphate–protein kinase G (GC-cGMP-PKG) pathway, as this kinase was also shown to phosphorylate the channel and then inhibit it [42]. Thus, more experiments with blockers of this pathway are necessary to obtain new insights into this issue.

Recently, Francis et al. have described an anti-arrhythmic effect of GPER activation on OVX guinea pigs, without changes in I_CaL_ [43]. The difference with the present work, in addition to the animal model, is the time for which the cells were exposed to G1. In Francis’s paper, the cells were exposed for 2 h to G1 1 μmol/L. It might be possible that our time exposition was sufficient to induce a rapid effect of GPER; moreover, after this time, this effect disappeared due to modifications of GPER expression in the cell membrane by desensitization, like that which occurs in the regulation of other G-protein-coupled receptors [44,45,46].

Finally, PI3K was another actor on the pathway studied. We employed wortmannin to block PI3K and, thus, the subsequent AKT phosphorylation and activation. Consistent with our hypothesis, wortmannin blocked the effects of G1 on calcium transients, contractility, and calcium currents. Accordingly, we demonstrated in a previous work that 1 μmol/L G1 augmented the phosphorylation level of AKT, and this effect was prevented with the GPER antagonist G15, using the myocardium of 3-month-old rats, the same age and species of the animals used in the present study [14]. Furthermore, several reports have described similar results regarding the activation of GPER and the PI3K-AKT pathway [10,37]. It is known that the selective activation of GPER and the phosphorylation of AKT increase NOS activity [13,37,47]; therefore, we propose to evaluate if this sequence of events is triggered after the activation of GPER. Thus, we determined NO production using DAF-FM-diacetate with or without wortmannin. G1 increased NO production, and the inhibition of the PI3K/AKT pathway prevented this effect. These results indicate that the pathway triggered by GPER activation is PI3K/AKT/NOS/NO.

Overall, we conclude that the selective activation of GPER decreases CaT amplitude and consequently cardiomyocyte contractility through an I_CaL_ reduction in Wistar male rats. In summary, we are reporting herein that the GPER-activated PI3K/AKT/NOS/NO pathway decreased LTCC activity and altered the contractile mechanism of the cardiomyocyte (Figure 8). Further experiments are needed to elucidate how NO specifically decreases I_CaL_.

## 4. Materials and Methods

### 4.1. Animals

All experiments were performed following the Guide for Care and Use of Laboratory Animals (NIH Publication No. 85-23, revised 2011) and approved by the Comité Institucional para el Cuidado y Uso de Animales de Laboratorio (CICUAL, No P02-03-2021) of the Faculty of Medicine, La Plata, Buenos Aires, Argentina. Male Wistar rats of 12-week-old were used for all experiments. The animals were raised in our vivarium and exposed to 12 h of light and 12 h of darkness, kept at 22 ± 2 °C until the sacrifice.

### 4.2. Ventricular Myocyte Isolation

Ventricular myocytes were isolated according to the technique previously described [48] with some modifications. The animals were anesthetized with Urethane (120 mg/100 g, Sigma Aldrich, St. Louis, MO, USA). The heart was attached via the ascendent aorta to a cannula, excised, and quickly mounted in a Langendorff apparatus. Then, a retrograde perfused at 37 °C with perfusion buffer (P-B) of the following composition (in mmol/L) was used: 146.2 NaCl, 4.7 KCl, 1 CaCl_2_, 10 HEPES, 0.35 NaH_2_PO_4_, 1 MgSO_4_, and 11 glucose (pH adjusted to 7.4 with NaOH), allowing the coronary arteries to be cleaned of residual blood. After a stabilization period of 4 min, the perfusion was switched to a nominally Ca^2+^-free P-B (P-B plus 20 μmol/L EGTA) for 4–5 min; this step allows the cardiac tissue to relax for better perfusion of the coronary arteries. The hearts were then perfused with collagenase type-II (140 units/mL, Worthington Biochemical) in P-B containing 50 μmol/L of CaCl_2_ for 12–15 min or until it became flaccid. All solutions were continuously bubbled with 100% O_2_. Finally, the heart was removed from the perfusion apparatus by cutting at the atrioventricular junction. The desegregated myocytes were separated from the undigested tissue and the CaCl_2_ concentration of perfusion buffer was increased in 5 steps to 1 mmol/L at room temperature (22–25 °C).

### 4.3. Measure of Calcium Transient and Cell Shortening

Calcium transient recordings were performed as previously described [49]. Briefly, isolated myocytes were loaded with 3 μmol/L Fura-2-AM (Thermo Fisher Scientific, Waltham, MA, USA) diluted in P-B for 10 min in the lightless chamber at room temperature. After a wash of the exceeding fluorescent dye with P-B, Fura-2 fluorescence kept in the myocytes was measured on an inverted microscope adapted for epifluorescence using IonOptix (IonOptix, Westwood, MA, USA). The cells were continuously perfused at a flow rate of 1 mL/min and stimulated via 2-platinum electrodes on either side of the perfusion chamber at 0.5 Hz. The perfused solution is equal to the P-B described previously (see above). The ratio of the Fura-2 fluorescence (at 510 nm) obtained after exciting the dye at 340 and 380 nm was taken as an index of intracellular Ca^2+^ (Ca^2+^_i_). Cell shortening was detected as sarcomere shortening (SS) using video-based motion detection in a region of interest (ROI), allowing the sarcomere detection without losing the cell in each shortening. All data were stored using software for an offline analysis using IonWizard Software version 6.3 (IonOptix). For sarcoplasmic-reticulum (SR) Ca^2+^ content measurement, a solution of P-B containing 15 mmol/L caffeine was rapidly applied to cells. The amplitude of the caffeine-induced Ca^2+^ transient was used to estimate SR Ca^2+^ content.

### 4.4. Patch Clamp Recording

L-type calcium current (I_CaL_) was recorded with a whole-cell patch clamp technique in voltage-clamp mode. An AxoPatch 200B amplifier and analog-to-digital converter Digidata 1322A (Molecular Devices, San Jose, CA, USA) were used to acquire I_CaL_ recorded with Clampex 9.2 Software (Molecular Devices). The current was filtered at 2 kHz. Borosilicate patch pipettes were made using a P-97 puller (Sutter Instruments, Novato, CA, USA) to a final resistance of 2.5–3.5 MΩ when filled with the pipette solution containing (in mmol/L) 135 CsCl, 1 MgCl_2_, 4 Na_2_ATP, 5 EGTA, and 10 HEPES (pH 7.2 with CsOH). Isolated rat ventricular myocytes were placed in a perfusion chamber and perfused at a flow rate of 1 mL/min with bath solution (HEPES Buffer) containing (in mmol/L) 5 CsCl, 122 NaCl, 1 MgCl2, 1 CaCl_2_, 10 tetraethylammonium chloride, 5 4-aminopyridine, 5 glucose, and 10 HEPES (pH 7.4 with NaOH). After achieving the whole-cell configuration, we waited 8–10 min for the stabilization of I_CaL_ and to discard rundown events. Cardiomyocytes were depolarized from a holding potential of −80 to −40 mV for 200 ms to inactivate sodium current and then to different test potentials ranging from −50 to +70 mV in 10 mV increments for 500 ms, delivered at 0.1 Hz. To obtain the current density vs. voltage (IV) curves, the steady state current was subtracted from the peak inward current corresponding to each test pulse. The peak inward current was normalized to cell capacitance (pA/pF). The patch clamp data were processed and analyzed with ClampFit 10.3 (Molecular Device).

### 4.5. Nitric Oxide Production

Cardiac myocytes were loaded with 5 μmol/L DAF-FM-diacetate (Thermo Fisher Scientific) for 30 min in the lightless chamber at room temperature (after a wash of the exceeding fluorescent dye with P-B) and imaged via epifluorescence on a Zeis 410 inverted confocal microscope (LSM Tech, Etters, PA, USA). The perfusion chamber was continuously perfused at a flow rate of 1 mL/min and stimulated via 2-platinum electrodes on either side of the chamber at 0.5 Hz. Excitation at 488 nm was provided by an argon laser and emission was collected in a range of 500–550 nm. Photographs were taken every 30 s for 25 min. ImageJ software version 1.54f was used for the analysis. Results were expressed as a percentage of time zero and slope fitted with linear regression. Slopes were normalized using the difference between treated and control conditions.

### 4.6. Treatments

All experiments were performed at room temperature (20–25 °C) and with bath solution with or without: 0.01% DMSO or ethanol (Sigma Aldrich, solvents); 10 nM E2 (Invitrogen, Carlsbad, CA, USA); 1 μM G1 (GPER agonist, Cayman Chemicals, Ann Arbor, MI, USA), 1 μM G36 (GPER antagonist, Cayman Chemicals), 100 nM Wortmannin (Invitrogen) or 100 nM L-NAME (Sigma Aldrich). All drugs were perfused for 15 min after 10 min of preincubation with the control solution corresponding to each experiment.

### 4.7. Statistics

GraphPad Prism 8 (GraphPad, San Diego, CA, USA) was used for all statistics analyses. Data were expressed as means ± SEM. A Shapiro–Wilk test was used to test normality. Data were compared with paired Student *t*-test or paired *t*-test with Wilcoxon test if variances could not be assumed to be equal. A value of *p* < 0.05 was considered statistically significant.

## Figures and Tables

**Figure 1 ijms-25-08993-f001:**
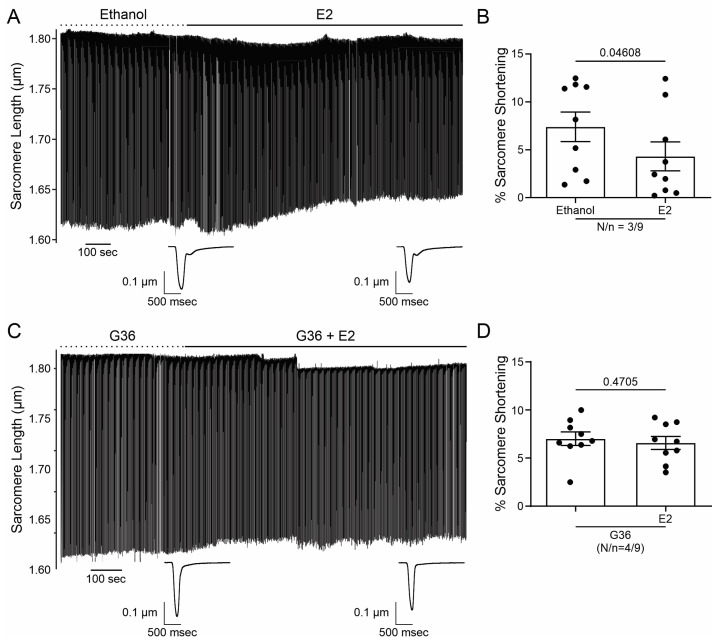
17-β-estradiol (E2) reduces cardiomyocyte contractility through G protein-coupled estrogen receptor (GPER) signaling. (**A**,**C**). Representative traces of cardiac cell contraction in the presence of E2 after ethanol or G36 plus E2 after G36 condition, respectively. Below are average traces of cell shortening at the end of each condition. (**B**,**D**). Average data of sarcomere shortening for each condition. *p* < 0.05 indicated a significant difference. N/n indicates animal and cell numbers, respectively.

**Figure 2 ijms-25-08993-f002:**
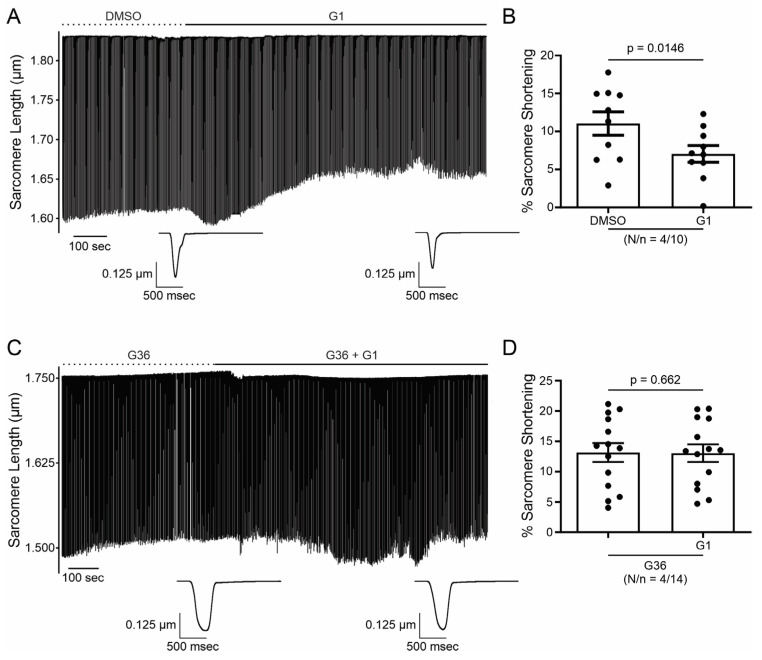
G1 reduces cardiomyocyte contractility through GPER signaling. (**A**,**C**). Representative traces of cardiac cell contraction in the presence of G1 after DMSO or G36 plus G1 after G36 condition, respectively. Below are average traces of cell shortening at the end of each condition. (**B**,**D**). Average data of sarcomere shortening for each condition. *p* < 0.05 indicated a significant difference. N/n indicates animal and cell numbers, respectively.

**Figure 3 ijms-25-08993-f003:**
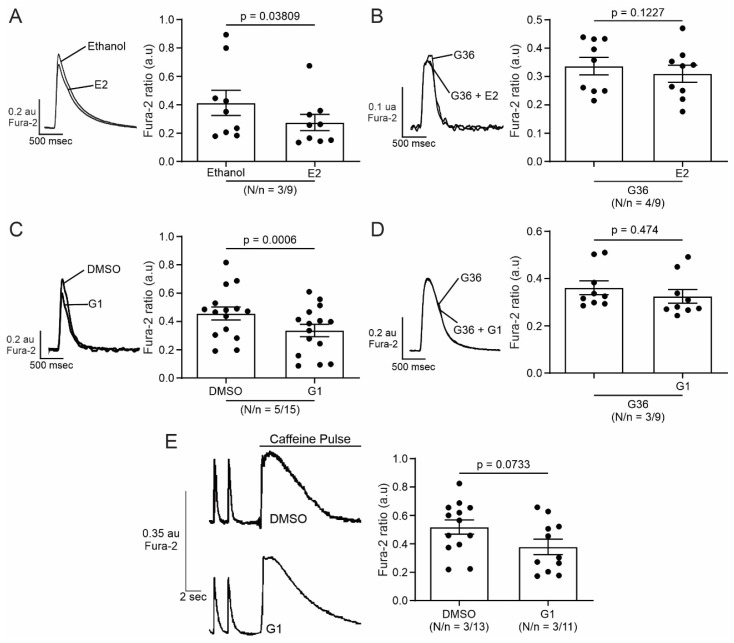
Activation of GPER with E2 or G1 decreases calcium transient in isolated cardiomyocytes, without changing sarcoplasmic reticulum (SR) calcium content. (**A**–**D**). Representative traces of calcium transient measured with Fura-2 in the presence of E2, G36 plus E2, G1, or G36 plus G1, respectively, after each control condition (left). The right panels represent average data of calcium transients for all the treatments applied. (**E**). Representative caffeine pulse-induced calcium transient (left) in the presence of DMSO (up) or G1 (down) condition. Right: representative average data of caffeine pulse-induced calcium transients. *p* < 0.05 indicates a significant difference. N/n indicates animal and cell numbers, respectively.

**Figure 4 ijms-25-08993-f004:**
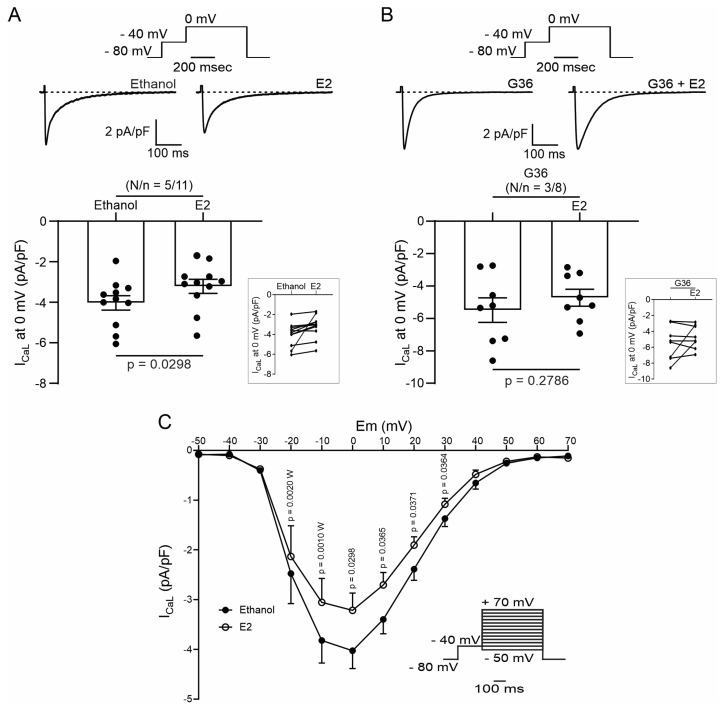
Activation of GPER with E2 decreases L-type calcium current (I_CaL_) in cardiomyocytes. (**A**,**B**). Top: voltage protocol and average traces of calcium current at 0 mV in the presence of E2 after ethanol or G36 plus E2 after G36 condition, respectively. Bottom: average data of calcium current at 0 mV and linked dot blot (inset) of each condition. (**C**). IV plot of I_CaL_ with ethanol or E2 treatment and voltage protocol (inset). *p* < 0.05 indicates a significant difference. N/n indicates animal and cell numbers, respectively.

**Figure 5 ijms-25-08993-f005:**
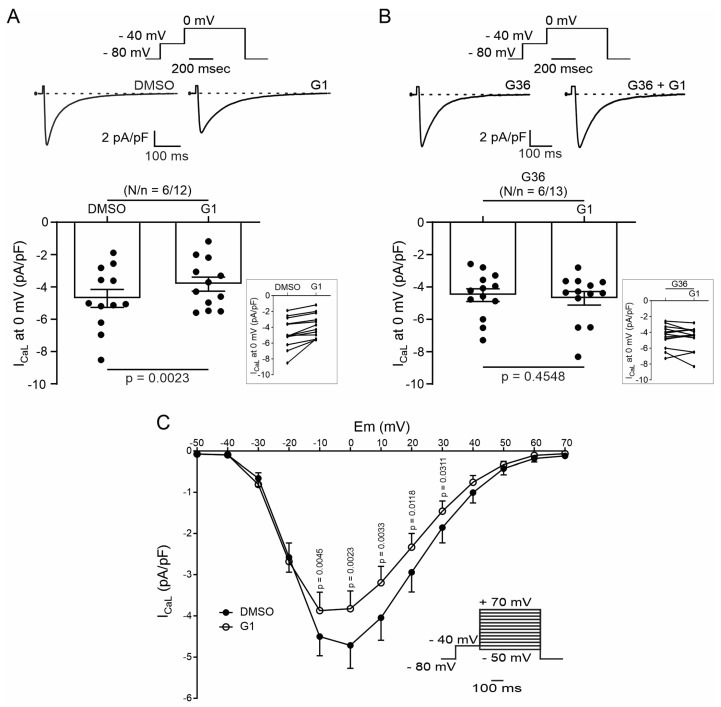
Activation of GPER with G1 decreases I_CaL_ in cardiomyocytes. (**A**,**B**). Top: voltage protocol and average traces of calcium current at 0 mV in the presence of G1 after DMSO or G36 plus G1 after G36 condition, respectively. Bottom: average data of calcium current at 0 mV and linked dot blot (inset) of each condition. (**C**). IV plot of I_CaL_ with DMSO or G1 treatment and voltage protocol (inset). *p* < 0.05 indicates a significant difference. N/n indicates animal and cell numbers, respectively.

**Figure 6 ijms-25-08993-f006:**
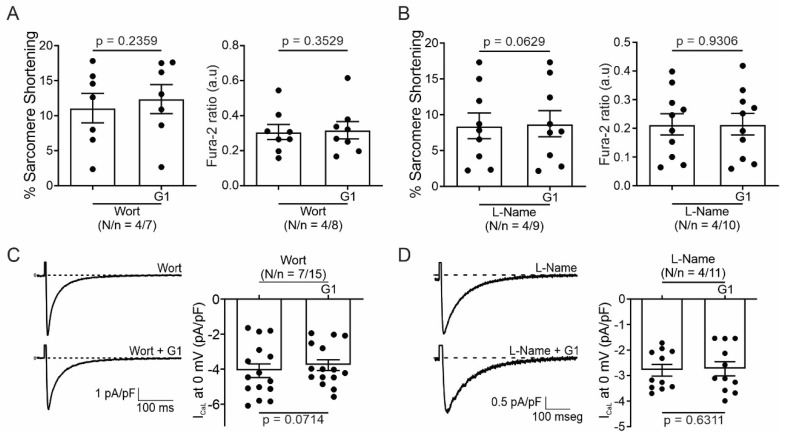
Phosphoinositide 3-kinase (PI3K) and nitric oxide synthase (NOS) are involved in the GPER-triggered intracellular pathway. (**A**,**B**). Average data of sarcomere shortening (left) and calcium transient (right) of cell treatment with wortmannin (PI3K blocker) or L-Name (NOS blocker) before exposing the cells to G1. (**C**,**D**). Average traces (left) and average data (right) of calcium current of cells treated with wortmannin (PI3K blocker) or L-Name (NOS blocker) before exposing the cells to G1. *p* < 0.05 indicates a significant difference. N/n indicates animal and cell numbers, respectively.

**Figure 7 ijms-25-08993-f007:**
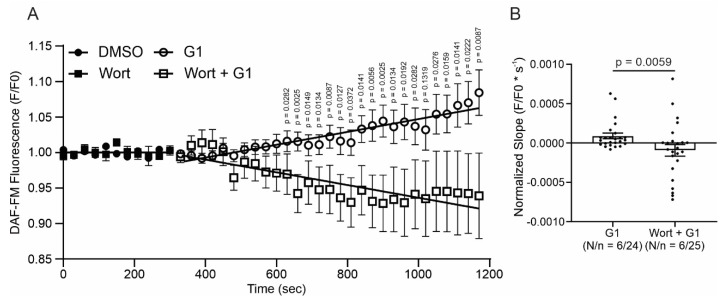
GPER increases nitric oxide (NO) production, activating PI3K before NOS activation. (**A**). DAF-FM fluorescence with respect to initial fluorescence of cells treated with G1 or wortmannin plus G1 after control condition (DMSO or wortmannin, respectively). (**B**). The average slope of G1 or wortmannin plus G1. *p* < 0.05 indicates a significant difference. N/n indicates animal and cell numbers, respectively. Since these values do not follow a normal distribution, Mann–Whitney test analysis was performed.

**Figure 8 ijms-25-08993-f008:**
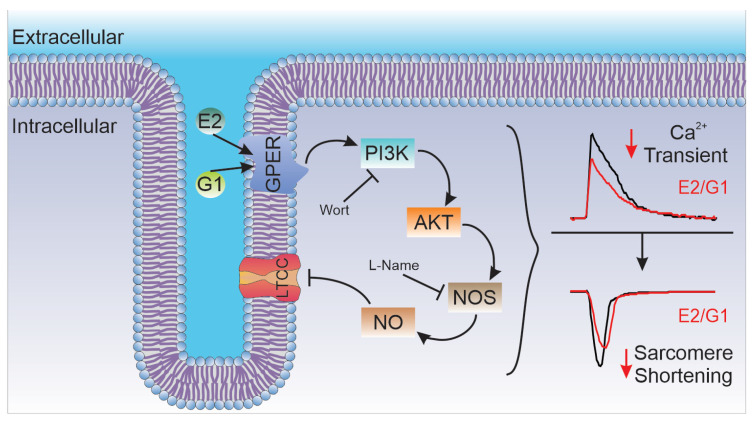
Schematic summary of the molecular events proposed to underlie GPER activation. Briefly, GPER activation with E2 or G1 activates PI3K, which stimulates NOS, possibly through the AKT pathway, and then increases myocardial NO production. NO would be responsible for L-type calcium current inhibition that impacts calcium transient and, therefore, cell shortening. E2, estradiol; G1, agonist of GPER; GPER, G-protein-coupled estrogen receptor; PI3K, phosphoinositide 3-kinase; AKT, protein kinase B; NOS, nitric oxide synthase; NO, nitric oxide; Wort, wortmannin, L-Name, NOS blocker; LTCC, L-type calcium channel. Full arrows indicate activation and incompletes arrows indicate inhibition.

## Data Availability

The original contributions presented in the study are included in the article, further inquiries can be directed to the corresponding author/s.

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
