# Peer review of "Activation of G Protein-Coupled Estrogen Receptor (GPER) Negatively Modulates Cardiac Excitation–Contraction Coupling (ECC) through the PI3K/NOS/NO Pathway"

_ijms, 2024, doi:10.3390/ijms25168993_

Round 1

Reviewer 1 Report

Comments and Suggestions for Authors

The estrogen hormone (17b-estradiol) signals via three different receptors: ER-alpha, ER-beta, and GPER. While the first two are intracellular receptors that once complexed with the hormone migrate into the nucleus acting as transcription factors, the latter is a member of the G-protein coupled receptor family.

GPER exerts quite a few effects at the levels of the heart, mostly beneficial. Indeed, GPER activation affords cardio-protection.

In the manuscript titled "Activation of G protein coupled estrogen receptors (GPER) negatively modulates cardiac excitation-contraction coupling (ECC) through the PI3K/NOS/NO pathway" Diaz-Zegarra L. and coworkers report the effect of activated GPER in the negative regulation of the cardiac excitation-contraction coupling. Moreover, they argue that the PI3K-NOS pathway mediates the action.

Overall, the manuscript offers some interesting hints though to some extent quite poor in findings.

Besides that, the authors did not plead sufficient experimental evidence to support their conclusions. Indeed, as such conclusions are rather weak. The authors are asked to provide biochemical evidence showing that PI3K downstream effectors are impaired, upon Wortmannin-dependent PI3K inhibition (i.e., a Western Blot of AKT/PKB phosphorylation status).

Section "Materials and Methods": the authors are asked either to significantly detail the part concerning the description of the sarcomere-shortening, or add a reference. Additionally, unless I missed something, it seems that throughout the manuscript details concerning the preincubation time are missing. 

Acronyms and abbreviations must be spelled out completely on initial appearance in the main text (e.g. E2 I guess 17b-estradiol).

The authors should carefully edit the whole section "Materials and Methods" because, according to iThenticate, currently it seems to be the result of a vigorous "cut and paste" process.  

A few typos are scattered throughout the main text (i.e., lines #117, #213, etc..).

Author Response

We thank the Reviewers for their comments that greatly helped to improve our work. The manuscript was modified following the suggestions of the Reviewers. The changes made in the manuscript were highlighted in yellow in the text.

Reviewer#1

The authors are asked to provide biochemical evidence showing that PI3K downstream effectors are impaired, upon Wortmannin-dependent PI3K inhibition (i.e., a Western Blot of AKT/PKB phosphorylation status).

-The comment of the Reviewer is appropriate. We demonstrated in a previous work that 1 mM G1 augments the phosphorylation level of AKT, and this effect was prevented with the GPER antagonist G15, using myocardium of 3-month-old rats, the same age and species of the animals used in the present work 1. A new paragraph explaining this issue was included in the present version of the manuscript (page 4, lines 191-196). In addition, other works also reported the involvement of wortmannin-sensitive AKT phosphorylation in GPER signalling (2-5), as quoted in the manuscript (page 4, line 162; page 4, lines 195-196). We thank the Reviewer for this comment.

Section "Materials and Methods": the authors are asked either to significantly detail the part concerning the description of the sarcomere-shortening or add a reference. Additionally, unless I missed something, it seems that throughout the manuscript details concerning the preincubation time are missing.

-The description of sarcomere-shortening measurement was included in Methods, page 6, line 254.

-The preincubation time with the drugs was now included in Methods, page 5, lines 242-245.

Acronyms and abbreviations must be spelled out completely on initial appearance in the main text (e.g. E2 I guess 17b-estradiol).

-The acronyms and abbreviations were spelled out completely on initial appearance in each section of the manuscript, as required by the Editorial. We thank the Reviewer for this comment.

The authors should carefully edit the whole section "Materials and Methods" because, according to iThenticate, currently it seems to be the result of a vigorous "cut and paste" process. 

-The whole Methods section was edited to avoid "cut and paste" process .

A few typos are scattered throughout the main text (i.e., lines #117, #213, etc..).

-Typos were corrected throughout the main text.

References

  1. De Giusti VC, Orlowski A, Ciancio MC, et al. Aldosterone stimulates the cardiac sodium/bicarbonate cotransporter via activation of the g protein-coupled receptor gpr30. J Mol Cell Cardiol. 2015;89(Pt B):260-267.
  2. Deschamps AM, Murphy E. Activation of a novel estrogen receptor, GPER, is cardioprotective in male and female rats. Am J Physiol Heart Circ Physiol. 2009;297(5):H1806-1813.
  3. Rocca C, Femmino S, Aquila G, et al. Notch1 Mediates Preconditioning Protection Induced by GPER in Normotensive and Hypertensive Female Rat Hearts. Front Physiol. 2018;9:521.
  4. Filice E, Recchia AG, Pellegrino D, et al. A new membrane G protein-coupled receptor (GPR30) is involved in the cardiac effects of 17beta-estradiol in the male rat. J Physiol Pharmacol. 2009;60(4):3-10.
  5. Lindsey SH, Cohen JA, Brosnihan KB, et al. Chronic treatment with the G protein-coupled receptor 30 agonist G-1 decreases blood pressure in ovariectomized mRen2.Lewis rats. Endocrinology. 2009;150(8):3753-3758.

Reviewer 2 Report

Comments and Suggestions for Authors

The manuscript written by Diaz-Zegarra et al. investigates how activation of the G protein-coupled estrogen receptor (GPER) affects cardioprotection. The results obtained with isolated cardiac myocytes show that activation of GPER with E2 or G-1 decreases intracellular calcium transients (CaT), sarcomere shortening (SS) and L-type calcium current (ICaL) and suggest that GPER activation induces a negative inotropic effect via the PI3K-NOS-NO pathway.

From a methodological point of view, the study is well conducted, although it could benefit from the determination of phospho-specific (active) and total PI3K isoforms by immunoblot analysis.

Minor points:

The use of abbreviations is not consistent – please write out all abbreviations in full the first time they are mentioned in the manuscript (e.g. the abbreviations used in line 45).

Line 126 ... pretreated with wortmannin (Figure 4 A and D) – the authors refer to the wrong figure, please correct this.

Lines 201-202 - approved by the Ethics Committee of the Faculty of Medicine, La Plata, Buenos Aires, Argentina - please add the Ethics Committee approval number.

Author Response

We thank the Reviewers for their comments that greatly helped to improve our work. The manuscript was modified following the suggestions of the Reviewers. The changes made in the manuscript were highlighted in yellow in the text.

Reviewer#2

The manuscript written by Diaz-Zegarra et al. investigates how activation of the G protein-coupled estrogen receptor (GPER) affects cardioprotection. The results obtained with isolated cardiac myocytes show that activation of GPER with E2 or G-1 decreases intracellular calcium transients (CaT), sarcomere shortening (SS) and L-type calcium current (ICaL) and suggest that GPER activation induces a negative inotropic effect via the PI3K-NOS-NO pathway.

From a methodological point of view, the study is well conducted, although it could benefit from the determination of phospho-specific (active) and total PI3K isoforms by immunoblot analysis.

-The comment of the Reviewer is appropriate. We demonstrated in a previous work that 1 mM G1 augments the phosphorylation level of AKT, and this effect was prevented with the GPER antagonist G15, using myocardium of 3-month-old rats, the same age and species of the animals used in the present work 1. A new paragraph explaining this issue was included in the present version of the manuscript (page 4, lines 191-196). In addition, other works also reported the involvement of wortmannin-sensitive AKT phosphorylation in GPER signalling (2-5), as quoted in the manuscript (page 4, line 162; page 4, lines 195-196). We thank the Reviewer for this comment.

Minor points:

The use of abbreviations is not consistent – please write out all abbreviations in full the first time they are mentioned in the manuscript (e.g. the abbreviations used in line 45).

-The acronyms and abbreviations were spelled out completely on initial appearance in each section of the manuscript, as required by the Editorial. We thank the Reviewer for this comment.

Line 126 ... pretreated with wortmannin (Figure 4 A and D) – the authors refer to the wrong figure, please correct this.

-The Reviewer is right. We apologize for this issue, which was corrected in the new version of the manuscript (page 3, line 128).

Lines 201-202 - approved by the Ethics Committee of the Faculty of Medicine, La Plata, Buenos Aires, Argentina - please add the Ethics Committee approval number.

-The Ethics Committee name and number was added to the manuscript (page 5, line 210-211).

References

  1. De Giusti VC, Orlowski A, Ciancio MC, et al. Aldosterone stimulates the cardiac sodium/bicarbonate cotransporter via activation of the g protein-coupled receptor gpr30. J Mol Cell Cardiol. 2015;89(Pt B):260-267.
  2. Deschamps AM, Murphy E. Activation of a novel estrogen receptor, GPER, is cardioprotective in male and female rats. Am J Physiol Heart Circ Physiol. 2009;297(5):H1806-1813.
  3. Rocca C, Femmino S, Aquila G, et al. Notch1 Mediates Preconditioning Protection Induced by GPER in Normotensive and Hypertensive Female Rat Hearts. Front Physiol. 2018;9:521.
  4. Filice E, Recchia AG, Pellegrino D, et al. A new membrane G protein-coupled receptor (GPR30) is involved in the cardiac effects of 17beta-estradiol in the male rat. J Physiol Pharmacol. 2009;60(4):3-10.
  5. Lindsey SH, Cohen JA, Brosnihan KB, et al. Chronic treatment with the G protein-coupled receptor 30 agonist G-1 decreases blood pressure in ovariectomized mRen2.Lewis rats. Endocrinology. 2009;150(8):3753-3758.

Round 2

Reviewer 1 Report

Comments and Suggestions for Authors

Overall, in the revised version of the manuscript titled "Activation of G protein coupled estrogen receptors (GPER) negatively modulates cardiac excitation-contraction coupling (ECC) through the PI3K/NOS/NO pathway" the authors managed to address most of the concerns. The reviewer thanks the authors for their efforts.